# Analysis of the Presence and Levels of IgG Antibodies Directed against the S1 Protein Receptor Binding Domain and the N Protein of SARS-CoV-2 in Patients with Multiple Sclerosis Treated with Immunomodulatory Therapies

**DOI:** 10.3390/vaccines12030255

**Published:** 2024-02-29

**Authors:** Joanna Kulikowska, Katarzyna Kapica-Topczewska, Monika Gudowska-Sawczuk, Agnieszka Kulczyńska-Przybik, Marcin Bazylewicz, Anna Mirończuk, Agata Czarnowska, Waldemar Brola, Barbara Mroczko, Jan Kochanowicz, Alina Kułakowska

**Affiliations:** 1Department of Neurology, Medical University of Bialystok, ul. Marii Skłodowskiej-Curie 24A, 15-276 Bialystok, Poland; katarzyna-kapica@wp.pl (K.K.-T.); grandholy@gmail.com (M.B.); anna-mironczuk@wp.pl (A.M.); agata.czarnowska@umb.edu.pl (A.C.); jan.kochanowicz@uskwb.pl (J.K.); alakul@umb.edu.pl (A.K.); 2Department of Neurodegeneration Diagnostics, Medical University of Bialystok, ul. Waszyngtona 15A, 15-269 Bialystok, Poland; monika.gudowska-sawczuk@umb.edu.pl (M.G.-S.); mroczko@umb.edu.pl (B.M.); 3Collegium Medicum, Jan Kochanowski University, Al. IX WiekówKielce 19, 25-317 Kielce, Poland; wbrola@wp.pl

**Keywords:** COVID-19, SARS-CoV-2, receptor binding domain, spike protein, nucleocapsid protein, multiple sclerosis, disease-modifying therapies, antibodies, vaccines, serology

## Abstract

The coronavirus 2019 disease (COVID-19) course and serological statuses of patients with relapsing–remitting multiple sclerosis (RRMS), treated with disease-modifying therapies (DMTs) are generally parallel that of the general population. Over the pandemic’s course, however, a notable increase in the number of RRMS patients who received vaccination against severe acute respiratory coronavirus 2 (SARS-CoV-2) and those who had COVID-19 (symptomatic and asymptomatic) was reported. This virus and/or vaccination likely influenced DMT-treated RRMS patients’ serological statuses regarding the presence of SARS-CoV-2 antibodies and their quantitative expression. This investigation assesses the presence and levels of the antibody directed against the S1 protein receptor binding domain (SRBD) and against the N protein of SARS-CoV-2 in 38 DMT-treated RRMS patients. The findings indicate that people vaccinated against SARS-CoV-2 exhibited significantly higher levels of IgG antibodies against S1-RBD at both assessment points. Patients with a prior history of COVID-19 demonstrated statistically significant increases in anti-N antibodies at visit 1, whereas such statistical significance was not observed at visit 2. DMT-treated RRMS patients generated neutralizing antibodies following vaccination and/or COVID-19 infection. Nevertheless, it is noteworthy that antibody levels more accurately reflect the serological status and exhibit a stronger correlation with vaccination than just the presence of antibodies.

## 1. Introduction

Coronavirus 2019 (COVID-19) is a disease caused by the severe respiratory syndrome coronavirus 2 (SARS-CoV-2) virus, which is responsible for the pandemic that started in 2019 [1]. The genetic material of the virus comprises a single-stranded RNA that encodes 16 non-structural proteins and four structural proteins: spike (S), nucleocapsid (N), envelope (E), and membrane (M) [2]. In a clinical context, the pivotal protein is the S protein, which was found to be accountable for facilitating virus entrance into the host cell via binding to the ACE2 (angiotensin-converting enzyme 2) receptor [3]. The spike protein was selected as a therapeutic target in the design of vaccines against the SARS-CoV-2 virus [4]. In addition, the antibodies against the S protein, which are produced as a result of active immunization (natural infection) and passive immunization (vaccination), are the only ones that have a neutralizing capability, thereby conferring protection against infection or reinfection [5]. It is also known that antibodies directed against the receptor binding domain (RBD) within the S1 subunit (anti-S1RBD antibodies) have the highest neutralizing capacity [6]. Moreover, the S1-RBD subunit exhibits minimal amino acid sequence homology compared to other coronaviruses [4]. The second clinically significant protein of the SARS-CoV-2 virus is the nucleocapsid (N) protein, which is responsible for the replication and transcription of viral RNA [7]. The N protein, as with the S protein, induces a humoral response. However, antibodies directed against the N protein are solely generated after natural infection but not after vaccination [8]. As in most European countries, in Poland, in December 2020, a mass vaccination program against COVID-19 began, initially available to selected risk groups (including healthcare workers). From May 2021, all adult Poles could receive the first dose of the vaccine. Registration for the second dose of the vaccine opened in November 2021. From April 2022, the second booster dose of the vaccine could be administered to people over 80 years of age, and from September 2022, all people over 12 years of age. The first available vaccine was the Comirnaty mRNA vaccine (Pfizer-BioNTech; Marburg, Germany). Subsequently, another mRNA vaccine, Spikevax (Moderna Biotech Spain, S.L, Madrit, Spain), was approved. Other vaccines approved in EU countries (including those used en masse in Poland) were vector vaccines: Vaxzeveria (AstraZeneca; Cambridge, UK) and Janssen Vaccine (Janssen-Cilag International NV; Beerse, Belgium) and a protein vaccine (Nouvaxovid Novavax; Gaithersburg, MD, USA). In May 2021, the Polish Neurological Society published an official position recommending COVID-19 vaccination for patients suffering from multiple sclerosis. Patients treated with beta interferons (INF), glatiramer acetate (GA), teriflunomide (TFN), dimethyl fumarate (DMF), and natalizumab (NTZ) should consider vaccination at every stage of treatment (no change in the therapy schedule is necessary). The guidelines specify groups of patients treated with fingolimod, ocrelizumab, cladribine, and alemtuzumab, where vaccination schemes are proposed depending on the time of DMT administration [9]. Insubsequent stages of the pandemic, the Polish Neurological Society updated its position on additional doses and booster doses [10].

Multiple sclerosis (MS) is a demyelinating autoimmune disease that impacts individuals across various age groups. The disease is predominantly diagnosed during the third decade of life [11]. MS is treated with immunomodulating and immunosuppressive drugs, termed disease-modifying therapies (DMTs). In Poland, 15 DMTs are currently available and financed by the National Health Fund. Nevertheless, the predominant cohort comprises individuals primarily undergoing treatment with dimethyl fumarate (DMF), glatiramer acetate (GA), or beta-interferon (INF) [12]. During the first months of the pandemic, physicians and patients wanted to know whether the treatment and the disease itself would negatively affect the course of SARS-CoV-2 infection [13]. In line with current knowledge, MS patients treated with most of the DMTs were infected by the SARS-CoV-2 infection at similar rates as the rest of society [14]. In addition, the response of this group of patients to vaccination against COVID-19,except for patients treated with anti-CD20 therapies and fingolimod, is normal [15]. At the time of the pandemic and afterward, the number of patients who had received subsequent vaccine doses and those who had come into contact with the SARS-CoV-2 virus increased. This increase probably affected the serological status of this specific group of patients, not only in terms of the mere presence of antibodies against S and N proteins but also the levels of these proteins, which seems to be important for future monitoring of the immunity of patients with MS who are treated with DMTs.

## 2. Materials and Methods

### 2.1. Study Group

The study group consisted of patients (*n* = 38) with relapsing–remitting multiple sclerosis (RRMS) who were treated with selected DMTs: DMF (36.84%; *n* = 14), GA (26.32%; *n* = 10), or INF (36.84%; *n* = 14). All examined patients were diagnosed in accordance with the McDonald criteria 2017 and were under the care of the Department of Neurology, Medical University of Bialystok [16]. Blood samples were collected twice between December 2021 and February 2023 (median 18.33 months) from each patient. During each visit, the following data were collected: (1) patient’s age, type and duration of disease-modifying drug used, COVID-19 vaccinations received (number of doses, dates of vaccinations, types of vaccinations), and documented positive result based on the polymerase chain reaction (PCR/COVID-19 antigen test). During both visits, the patients were examined by a neurologist. All individuals signed informed consent to participate in this study.

A total of 57.89% (*n* = 22) of the study group were women. The average age at the first visit was 44.5 years. The average duration of the disease was 9 years. The average duration of using DMF is 3.4 years (SD ± 1.4–6.3), GA 5.1 years (SD ± 1.3–9.1), INF 10 years (SD ± 1.0–9.2). Among all vaccine doses received by patients, 75% were vaccinated with the Comirnaty vaccine (Pfizer-BioNtech; Marburg, Germany)., 12.50% with the Vaxzeveria vaccines (AstraZeneca; Cambridge, Great Britain, 6.94% with the Spikevax (Moderna Biotech Spain, S.L, Madrit, Spain) and 5.56% with the Janssen Vaccine (Janssen-Cilag International NV; Beerse, Belgium).

The study was approved (approval NAPK.002.230.2020) by the Bioethics Committee at the Medical University of Bialystok, Poland.

The detailed clinical characteristics of the study group are shown in Table 1 and Table 2.

### 2.2. Laboratory Tests

An assessment of antibodies against the SARS-CoV-2 virus was conducted: (1)IgG antibodies against the receptor binding domain of S1protein (IgG-S1RBD) and (2) IgG antibodies against N protein (IgG-N). Serum levels of the IgG-S1RBD and IgG-N antibodies were measured by chemiluminescent microparticle immunoassay (CMIA) according to the manufacturer’s instructions. The result in the chemiluminescent reaction was assessed as relative light units (RLU) using the automatic Alinity system (Abbott, Chicago, IL, USA) according to the manufacturer’s instructions. The level of serum antibodies was directly proportional to the RLU detected by the system optics. The S/C (serum/cut-off) index was determined based on the above relationship. A titer ≥ 1.4 (IgG-N) and ≥50 (IgG-SRBD) was considered a positive result.

### 2.3. Statistical Analysis

The statistical analysis was based on a description of groups of patients classified by DMT and survey data (sex, age, COVID-19 status, vaccination status). The significance level of the statistical tests in this analysis was set at α = 0.05. The normality of the distributions of the variables was analyzed using the Shapiro–Wilk test. Numerical variables with distributions deviating from the normal distribution were reported as *Mdn* (*Q1*, *Q3*). Examination of differences within a numerical variable with a non-normal distribution between two groups was performed with the Wilcoxon rank sum test and between three or more groups was performed with the Kruskal–Wallis rank sum test. The significance of differences between pairs of groups was tested using Dunn’s test. The effects of vaccination or pastCOVID-19 infection over time (visits 1 and 2) on the concentration of SARS-CoV-2 IgG (S-RBD, N) were examined using a linear mixed model. In the case of dichotomous response variables (SARS-CoV-2 IgG positive result for S-RBD or N), a generalized linear model was applied. The magnitude of the effect between categories within an exploratory variable with more than two categories (such as the number of vaccine doses) was estimated by contrast analysis of the estimated marginal means with the Tukey adjustment. Spearman’s rank correlation coefficient (rho) was used to measure the strength and direction of association between two variables. Analyses were conducted using the R Statistical language (version 4.1.1; R Core Team, 2021) on Windows 10 x64 (build 19045).

## 3. Results

### 3.1. Analysis of Antibodies against S1 Protein

#### 3.1.1. Impact of Vaccination

At visit 1, patients with RRMS who had not been vaccinated against SARS-CoV-2 accounted for 39.47% (*n* = 15) and vaccinated 60.53% (*n* = 23). At visit 1, among unvaccinated patients with RRMS, 33.33% (*n* = 5) had no antibodies against S1-RBD, while 66.67% (*n* = 10) had positive antibodies. Among vaccinated patients with RRMS, 91.30% (*n* = 21) showed positive anti-S1RBD IgG antibodies. At visit 2, 28.95% (*n* = 11) of RRMS patients were unvaccinated, and 71.05% (*n* = 27) were vaccinated against SARS-CoV-2. Among the unvaccinated patients, 9.09% (*n* = 1) tested negative for S1RBD antibodies, while 90.91% (*n* = 10) tested positive. Of the vaccinated patients, 3.70% (*n* = 1) were S1RBD negative and 96.30% (*n* = 26) positive. Statistical analysis showed that percentages of IgGS1RBD results between vaccinated and unvaccinated patients with RRMS were not statistically significant (visit 1: *p* = 0.089; visit 2 *p* = 0.501). However, at visit 1, the number of positives was higher in the vaccinated group than in the unvaccinated group, which was significant at the trend level (0.050 ≤ *p* < 0.100). In addition, a significant main effect of the time factor (the odds of getting a positive SARS-CoV-2 anti-S1RBD result at visit 2 was significantly higher (41.227-fold) than at visit 1. Detailed data concerning the presence of anti-S1RBD antibodies are shown in Table 3.

At visit 1, the mean anti-S1RBD antibody level was 16,863.40 AU/mL among vaccinated patients with RRMS and 197.90 AU/mL among unvaccinated. At visit 2, the mean level of anti-S1-RBD antibodies was 6997.30 AU/mL among vaccinated patients with RRMS and 1342.50 AU/mL among unvaccinated. The statistical analysis showed that vaccination had a statistically significant effect on anti-S1-RBD antibody levels at visit 1 (*p* < 0.001) and visit 2 (*p* = 0.038). Detailed data on the level of anti-S1RBD antibodies are given in Figure 1.

#### 3.1.2. Impact of COVID-19

A positive antigen test or PCR was a COVID-19 infection indicator. At visit 1, patients withCOVID-19 (+), 100% (*n* = 7) were positive for anti-S1RBD antibodies. However, among patients withCOVID-19 (−), 77.42% (*n* = 24) were positive for anti-S1RBD antibodies. At visit 2, amongCOVID-19 (+) patients, 100% (*n* = 9) tested positive for anti-S1RBD antibodies. However, among COVID-19 (−) patients, 93.10% (*n* = 27) tested positive for anti-S1RBD antibodies. Detailed data on the presence of anti-SRBD antibodies are shown in Table 4. The statistical analysis showed that the percentages of anti-S1RBD results between patients with COVID-19 (+) and COVID-19 (−) illness were not statistically significant (visit 1: *p* = 0.309; visit 2 *p* = 1.00).

History of COVID-19 registered positive polymerase chain reaction (PCR)/antigen test in the past. Among the patients with confirmed COVID-19, just one patient was hospitalized due to COVID-19, and the patient received convalescent plasma and steroids.

At visit 1, among COVID-19 (+) patients, the mean level of anti-S1RBD antibodies was 27,086.20 AU/mL, while among COVID-19 (−) patients, it was 1953.90 AU/mL. At visit 2, among COVID-19 (+) patients, the level of anti-S1RBD antibodies was 3886.90 AU/mL, while among COVID-19 (−) patients, it was 4165.20 AU/mL. Statistical analysis showed that COVID-19 survivors had statistically significantly higher levels of anti-S1RBD antibodies at visit 1 (*p* = 0.001) but not at visit 2 (*p* = 0.410). Detailed data on the level of anti-S1RBD antibodies are presented in Figure 2.

### 3.2. Analysis of Antibodies against N Protein

At Visit 1, 28.57% (*n* = 2) of COVID-19 (+) patients were positive for anti-N antibodies. However, among COVID-19 (−) patients, 9.68% (*n* = 3) were negative for anti-N antibodies. At visit 2, 66.67% (*n* = 6) of COVID-19 (+) patients were positive for anti-N antibodies. Among COVID-19 (−) patients, 37.93% (*n* = 11) tested positive for anti-N protein antibodies. Detailed results of anti-N protein antibodies are shown in Table 5. Statistical analysis showed that the percentages of IgG-N results between patients withCOVID-19 (+) and COVID-19 (−) were not statistically significant (first visit: *p* = 0.223; second visit *p* = 0.249).

At visit 1, among COVID-19 (+) patients, the mean level of IgG-N was 0.90 AU/mL, while among COVID-19 (−) patients, it was 0.14 AU/mL. At visit 2, amongCOVID-19 (+) patients, the level of anti-N antibodies was 1.99 AU/mL, while for COVID-19 (−) patients, it was 0.79 AU/mL. Detailed data on the level of anti-N antibodies are shown in Figure 3. The statistical analysis showed that COVID-19 (+) patients had statistically significantly higher levels of anti-N antibodies at visit 1 (*p* = 0.040) but not at visit 2 (*p* = 0.363). A significant main effect of time was observed, indicating that anti-N levels were significantly higher than at visit 1. Levels of antibodies IgG-S1RBD and IgG-N according to particular DMTs are presented in the Appendix A.

## 4. Discussion

Our study shows the result of the analysis of both the presence and actual levels of antibodies directed against the receptor binding domain of the S1 protein and against the N protein of the SARS-CoV-2 virus in patients with RRMS who were treated with DMF, GA, or INF. During the first months of the pandemic, it was unknown whether patients undergoing immunomodulatory treatment were at risk of a more severe course of the disease [17]. Current clinical experience shows that SARS-CoV-2 infection can impact on central nervous system, but patients with MS, in most cases, do not suffer more seriously from COVID-19 than the general population [14]. Risk factors for a more severe course, such as male sex, comorbidities, or severe disability, are similar to those in the non-MS group [18,19,20]. The subsequent months of the pandemic also showed that patients treated with selected and often used DMTs showed an adequate humoral response to vaccinations [21,22]. Moreover, the side effects of vaccinations against SARS-CoV-2 were mild, and the vaccinations themselves were safe in this group of patients [14,23]. During the first period of the pandemic, the serological status of patients was also assessed. It was shown that vaccinations significantly induced the production of neutralizing antibodies in patients treated with DMT, GA, or INF [24]. It is worth noting that it is currently known that in the group treated with anti-CD20 and sphingosine-1-phosphate modulators, the course of the disease may be more severe, and the immune response to vaccinations may be impaired [22]. Over the duration of the pandemic and after it ended, the group of patients who received subsequent doses of the vaccine, in addition to those who had COVID-19 infections, grew. For this reason, the serological status of this group of patients may change, and it seems that in addition to the presence of antibodies, their levels may also be important. Neutralizing IgG antibodies are known to increase from two to eight weeks post-infection, followed by a decline ranging from four to six months with a median time to seronegativity of approximately two years [25]. 

The literature shows that the most specific and the least likely to cross-react antibodies are those directed against the S1 protein receptor binding domain, so we tested these antibodies in our research [26]. At visit 1, the presence of these antibodies was found in almost 82% of the study group. Analyzing the subgroups at visit 1, we could see that in the vaccinated group, neutralizing antibodies were present in 91.30% of patients with RRMS. At visit 2, in the entire study group, neutralizing antibodies were found in 94.74% of patients, including 96.30% of vaccinated and 90.91% of unvaccinated patients. In the latter group of patients, the presence of neutralizing antibodies is probably due to passive immunization after asymptomatic contact with SARS-CoV-2. The results obtained in our study are similar to previously published studies. A meta-analysis by Gombolay et al. showed that the humoral response after vaccination occurs in 77% of patients with MS compared to 93% of the healthy population [22]. This study also analyzed individual DMTs and found that 96% of those were treated with INF, 95% of those treated with GA, and 99% of those treated with DMF [22]. However, results from the statistical analysis did not show a difference between vaccinated and unvaccinated people, which is probably due to the high prevalence of the SARS-CoV-2 virus in the population of patients with MS. In Poland, the OBSER-CO seroepidemiological study was conducted (IV series of analysis in 2021 and 2022) based on the WHO-Unity protocol: “Population-based age-stratified seroepidemiological investigation protocol for COVID-19 infection” [27]. Comparing the results obtained in our study to the results conducted as part of OBSER-CO in northeastern Poland on a group of patients ofa similar age, we can note that during visit 1, more patients with MS were vaccinated than in the general population (61% vs. 31 -58%). However, during visit 2, these proportions practically equalized (71% vs. 67%). Comparing the seroprevalence (presence of IgG-S antibodies) in the MS population to the general population in northeastern Poland during visit 1, it can be seen that the prevalence of neutralizing antibodies was much higher in the MS group (84.2% vs. 57–73%) [28]. This can be explained by the higher vaccination rate of the studied group of patients, as shown by previous data. During visit 2, within 1.5 years later, the seroprevalence was practically at the same level (94.7% vs. 93.4%). Data from OBSER-CO 2023 have not been published yet.

At visit 1, the number of neutralizing antibody positives was higher in the group vaccinated at the trend level (*p* = 0.089). Many of these patients treated with DMTs had asymptomatic contact with the virus, which induced the presence of antibodies. In the next step, levels of antibodies directed against the receptor binding domain of the S1 protein were analyzed. A statistical analysis of these data showed that vaccinated patients with RRMS had statistically significantly higher antibody levels at both visits than unvaccinated subjects. It is worth noting that in vaccinated people with multiple sclerosis (PwMS), neutralizing antibody levels were lower at visit 2 (16,863.40 AU versus 6997.30 AU at visit 1). A similar observation was made after analyzing IgG-S1RBD and IgG-N according to particular DMTs. At visit 2, the median time since the last vaccination was longer than at visit 1 (<2 months since the last vaccination at visit1 versus approximately 12–18 months at visit 2). Due to this strong induction of antibody levels, subsequent vaccination doses against COVID-19 are still highly recommended forMS patients. Interestingly, the level of neutralizing antibodies in unvaccinated patients increased at visit 2 (1342.5 AU) compared to visit 1 (197.80 AU) but did not reach the level observed in vaccinated PwMS. However, it should be noted that, currently, the level of neutralizing antibodies that would protect against COVID-19 has not been determined. It is not known whether a higher level clearly means higher protection. Research conducted by Hickey et al. on the general population showed that the levels of antibodies in vaccinated people were significantly higher than in people after infection. Studies show that in addition to antibody levels, avidity was higher in the vaccinated group, which may even better reflect the level of protection against reinfection. For each vaccine, circulating antibody levels decreased one to four months after the second dose [29]. It is worth emphasizing at this point that the protective level of neutralizing antibodies has not yet been determined. In a multicenter study, a group of 2nd and 3rd doses administered to patients with MS similarly caused a decrease in neutralizing antibodies within six months post-vaccination but still remained high compared to unvaccinated subjects. We also analyzed IgG antibodies against the N protein, which are induced only after natural contact with the virus. No significant differences in the percentage of positive results between people with and without previous COVID-19 were found. After analyzing the levels of antibodies, a statistically significantly higher level of antibodies was shown only during the first visit (2021). This seems to be related to less frequent testing of patients forSARS-CoV-2 during the subsequent years of the pandemic (second visit in 2023) and to the greater prevalence of the virus in the population. In addition, a significant factor is also the short duration of antibodies directed against the N protein, namely, less than one year. All these factors make the interpretation of the presence and levels of anti-N protein antibodies difficult and should be closely correlated with the clinical status of the patient. Literature shows that vaccinated people who contracted COVID-19 have higher levels of antibodies compared to people who were only vaccinated or only after natural infection (hybrid immunity) [30]. An interesting observation is that during visit 2, in the subgroup of patients with confirmed COVID-19 in the past, the level of neutralizing antibodies was lower than in the group of patients without documented COVID-19 (3886.9 AU versus 4165.2 AU). In addition to high vaccination rates in the subsequent years of the pandemic, it can be assumed that a significant percentage of the population and patients with MS have already had natural contact with the virus. Our data shows that none of the participants tested positive (PCR or antigen test) for COVID-19 between visits 1 and 2. Moreover, the levels of IgG-N at visit 2 were higher than that in visit 1 in vaccinated and unvaccinated patients and according to particular DMTs. That may indicate that part of the study group probably had SARS-CoV-2 infection and did not decide to test or had asymptomatic infections. Recent studies and literature reviews also point to the importance of assessing not only antibodies but also cell-mediated immunity, which seems to last longer than humoral immunity [31]. 

Our research has limitations, one of which includes a small study group. The small size probably contributed to the limitations of statistical analyses and did not allow for reliable analysis of particular DMT subgroups. In conclusion, our research shows that, in addition to the presence of antibodies against the S1 protein (RBD), it is important to assess their levels. Patients with RRMS who were vaccinated against SARS-CoV-2 had significantly higher levels of neutralizing antibodies in subsequent years of the pandemic. The assessment of anti-N antibodies is difficult due to the high seroprevalence of the virus in the population and the short half-life and should be closely correlated with the clinical picture.

## 5. Conclusions

Our research shows that SARS-CoV-2 vaccinated patients with RRMS treated with DMT, GA, or INF have statistically significantly higher levels of antibodies directed against the receptor binding domain of the S1 protein compared to unvaccinated. This was observed over the course of two years of the pandemic. Levels of neutralizing antibodies seem to better reflect the level of protection against the SARS-CoV-2 virus than their presence alone, but this requires further research. In the presented retrospective study, it was observed that patients treated with the selected DMTs (INF, GA, DMF) were immunocompetent in terms of the production of neutralizing antibodies. In conjunction with the above data and current world literature, recommendations for preventive vaccinations for MS patients are justified. Although a clearly protective level of antibodies has not been currently determined, a higher level potentially provides better protection for patients against disease and reinfection. Further randomized studies are still necessary.

## Figures and Tables

**Figure 1 vaccines-12-00255-f001:**
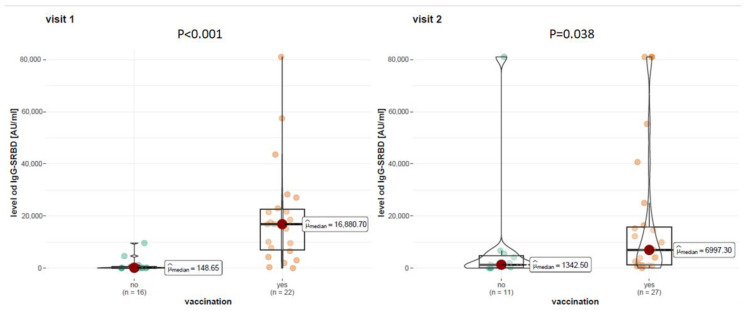
Level of anti-SRBD antibodies in COVID-19 vaccinated and unvaccinated patients at visit 1 and 2.

**Figure 2 vaccines-12-00255-f002:**
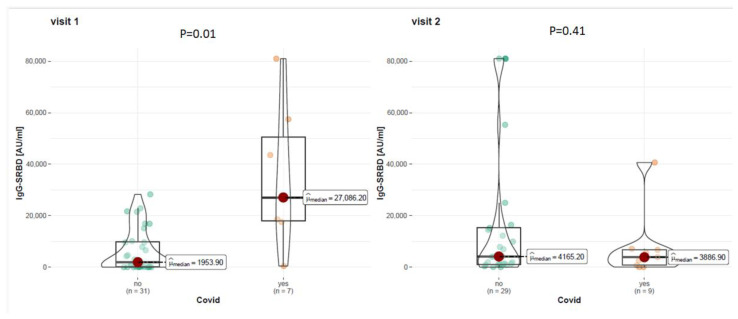
Level of anti-SRBD IgG antibodies in patients with or without registered history of COVID-19 at visit 1 and 2.

**Figure 3 vaccines-12-00255-f003:**
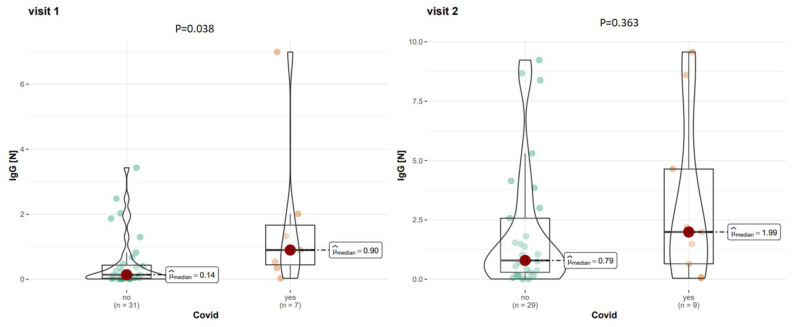
Level of IgG-N antibodies in patients with or without registered history of COVID-19 on visit 1 and 2.

**Table 1 vaccines-12-00255-t001:** Characteristics of the study group.

Sex	Female	57.89% (*n* = 22)
	Male	42.11% (*n* = 16)
Age (on visit 1)		44,50 (36.25, 48.75) ^1^
DMT	DMF	36.84% (*n* = 14)
	GA	26.32% (*n* = 10)
	INF	36.84% (*n* = 14)
Time between visit 1 and 2		18.33 (17.70, 18.84) ^1^

^1^ Median (Q1, Q3); DMT, dimethyl fumarate; GA, glatiramer acetate; INF, interferon beta.

**Table 2 vaccines-12-00255-t002:** Detailed characteristics of study groups.

		Visit 1	Visit 2
Vaccinated ^2^		60.52% (*n* = 23)	71.05% (*n* = 27)
	One dose	26.09% (*n* = 6)	3.70% (*n* = 1)
	Two doses	73.91% (*n* = 17)	40.74% (*n* = 11)
	Three doses	0	48.15% (*n* = 13)
	Four doses	0	7.41% (*n* = 2)
Time between first dose and visit [months]		1.69 (1.10; 2.48) ^1^	27.37(26.24; 28.21) ^1^
Time between second dose and visit [months]		1.03 (0.38; 1.53) ^1^	18.74 (17.77; 19.66) ^1^
Time between third dose and visit [months]		-	11.70 (10.81; 12.42) ^1^
Time between fourth dose and visit [months]		-	11.70 (1.81; 12.42)
unvaccinated		39.47% (*n* = 15)	28.95% (*n* = 11)
COVID-19“+” ^3^		18.42% (*n* = 7)	23.68% (*n* = 9)
COVID-19“−” ^4^		81.58% (*n* = 31)	76.32%(*n* = 29)
Time between COVID-19 and visit [months]		4.93 (4.70; 5.95) ^1^	23.56 (12.78; 25.00) ^1^

^1^ Median (Q1, Q3); ^2^ Vaccinated—people with multiple sclerosis (PwMS) vaccinated against coronavirus 2019 disease (COVID-19); ^3^ COVID-19“+”—PwMS with registered positive PCR/antigen test in the past; ^4^ COVID-19“−”—PwMS with no registered positive PCR/antigen test in the past.

**Table 3 vaccines-12-00255-t003:** Presence of anti-S1RBD IgG antibodies in COVID-19 vaccinated and unvaccinated patients at visits 1 and 2.

	IgG-S1RBD Results	Vaccination against SARS-CoV-2		*p*-Value ^1^
		No	Yes	
Visit 1 (*n* = 38)	Negative	33.33% (*n* = 5)	8.70% (*n* = 2)	*p* = 0.089
	Positive	66.67% (*n* = 10)	91.30% (*n* = 21)	
Visit 2 (*n* = 38)	Negative	9.09% (*n* = 1)	3.70% (*n* = 1)	*p* = 0.501
	Positive	90.91% (*n* = 10)	96.30% (*n* = 26)	

^1^ Fisher’s exact test.

**Table 4 vaccines-12-00255-t004:** Presence of anti-SRBD IgG antibodies in patients with or without a registered history of COVID-19 at visit 1 and visit 2.

	IgG-SRBD Results	History of COVID-19		*p*-Value ^1^
		No	Yes	
Visit 1 *n* = 38	Positive	77.42% (*n* = 24)	100% (*n* = 7)	*p* = 0.309
	Negative	22.58% (*n* = 7)	0.00%	
Visit 2 *n* = 38	Positive	93.10% (*n* = 27)	100% (*n* = 9)	*p* = 1.00
	Negative	6.90% (*n* = 2)	0.00%	

^1^ Fisher’s exact test.

**Table 5 vaccines-12-00255-t005:** Presence of IgG-N antibodies in patients with or without registered history of COVID-19 at visit 1 and visit 2.

	IgG-N Results	History of COVID-19		*p*-Value ^1^
		No	Yes	
Visit 1 *n* = 38	Positive	9.68% (*n* = 3)	71.43% (*n* = 35)	*p* = 0.223
	Negative	90.32% (*n* = 28)	28.57% (*n* = 2)	
Visit 2 *n* = 38	Positive	37.93% (*n* = 11)	66.67% (*n* = 6)	*p* = 0.249
	Negative	62.07% (*n* = 18)	33.33% (*n* = 3)	

^1^ Fisher’s exact test. History of COVID-19 registered positive PCR/antigen test in the past.

## Data Availability

Data are contained within the article and Appendix A.

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
