# Peer review of "Analysis of the Presence and Levels of IgG Antibodies Directed against the S1 Protein Receptor Binding Domain and the N Protein of SARS-CoV-2 in Patients with Multiple Sclerosis Treated with Immunomodulatory Therapies"

_vaccines, 2024, doi:10.3390/vaccines12030255_

Round 1

Reviewer 1 Report (Previous Reviewer 2)

Comments and Suggestions for Authors

In its present form, the article can be accepted for publication.

Author Response

Dear Reviewer 1,

We are very pleased, that the improvements we introduced satisfied the Reviewer. 

We are greateful for all valuabele comments.

The Authors

Reviewer 2 Report (Previous Reviewer 3)

Comments and Suggestions for Authors

Authors partially addressed my previous comments. The paper now was improved. 
my observations:

- difference between two groups should be assessed with Mann-Whitney U test;

- drug therapy such as antiviral can led to a lower IG response. They should be included in the analysis. The lack of this information can bias all results;

- the lack of a control group is the strongest limitation of the study.

Author Response

Dear Reviewer 2.

Thank you once again for taking the time to review our manuscript and for all your valuable comments.

We respond to the comments below:

- difference between two groups should be assessed with Mann-Whitney U test;

 We analyzed the statistical calculations once again. We examined intergroup differences using the Mann-Whitney U test, which is also called the "Wilcoxon rank sum test". We hope that the statistics we conducted are clearly explained in the manuscript. -  drug therapy such as antiviral can led to a lower IG response. They should be included in the analysis. The lack of this information can bias all results;

The research was carried out in the form of cross-sectional studies based on WHO protocols, carried out on the general population in many countries around the world, including Poland. Antiviral treatments were not routinely assessed in these protocols. Similarly, other published peer-reviewed articles included in the bibliography that described the MS patient population did not evaluate antiviral therapy. However, we appreciate the suggestion that this data would be of great value in serological analysis and we will certainly include this data in our future studies.

-  the lack of a control group is the strongest limitation of the study

We agree that the lack of a control group is the greatest limitation of the presented research. However, in our opinion, the fact that the OBSER-CO study (based on WHO protocols) was conducted at the same time and in a general population of similar age to our study group, to which we refer in the discussion, indicates that our results can be applied to the general population. Of course, we agree that in our future research we will assemble a control group ourselves. However, no seroepidemiological studies have been published so far on a group of patients with multiple sclerosis in north-eastern Poland, therefore we believe that the results we present are valuable.   

We hope that the answers we present will satisfy the Reviewer . Thank you once again for all your valuable comments and tips.

The Authors,

Reviewer 3 Report (Previous Reviewer 4)

Comments and Suggestions for Authors

the authors adequately modified their manuscript. It is now suitable for publication. 

Author Response

Dear Reviewer 3,

We are very pleased, that the improvements we introduced satisfied the Reviewer. 

We are greateful for all valuable comments.

The Authors

Round 2

Reviewer 2 Report (Previous Reviewer 3)

Comments and Suggestions for Authors

Observations reported by Authors about drug therapies are not enough. Antivirals and monoclonal antibodies are one of main therapeutic option for immunocompromised patients. Their use can influence IGG titer so they have to be considered. THe lack of this information reamain a strong limitation. Several studies from other countries considered these covariates in the analysis.

About the lack of  a control group, I can't find real similarities between the present study and OBSER-CO study. 

Author Response

Dear Reviewer,

According to your comments, we contacted the patients and analyzed the data ones again. Withing the patients with confirmed COVID-19, just one patients, treated with glatiramer acetate, was hospitalyzed due to COVID-19 and the patinets received convalescent plasma and steroids. No antiviral drugs or monoclonal antibodies were used. The hospitalization was in December 2020. First visit in our study with blood sample collection was in June 2021 and second visit in January 2023. In both invastigation the patients demonstrated high levels of IgG against N and S protein. The remaining patients from the study group did not received the antiviral drugs or monoclonal antibodies. We suplemented the manuscript with data on hospitalization and treatment of COVID-19 in study group.

We agree that the lack of a control group is the strongest limitation of the study. In OBSER-CO study seroepidemiological features such as IgG-S and IgG-N were investigated. The laboratory analysis and the protocol of the study were familiar in our opinion. Morover the study was conducted in every polish voivodeship (also Podlaskie) in particular age group.  We regret that the comparison with these studies did not satisfy the Reviewer but we hope that beside that, it will allow the manuscript to be forwarded to further stages of publication. There has been no published data so far on a group of parients with multiple sclerosis in north-eastern Poland, therefore we belive tah the results we presented are valuable.  

The Authors, 

Round 3

Reviewer 2 Report (Previous Reviewer 3)

Comments and Suggestions for Authors

Ok

This manuscript is a resubmission of an earlier submission. The following is a list of the peer review reports and author responses from that submission.

Round 1

Reviewer 1 Report

Comments and Suggestions for Authors

Authors assayed antibodies directed against two COVID-19 proteins (S and N) in a sample of 38 multiple sclerosis (MS) patients treated by disease-modifying treatments (DMT). DMT were mainly drugs known for their low immunosuppressive effect (interferon, glatiramer, and more or less fumarate). Two time points were used: shortly after the beginning of the vaccinal campaign, and about 2 years later.

As expected, vaccinated patients demonstrated a higher level of antibodies against S protein than unvaccinated or infected patients. This effect vanished at the second time point between infected and non infected. Levels of antibodies against N protein in patients with history COVID-19 infection, and in unvaccinated at time point 2, suggesting that many infections were non-symptomatic. 

In our opinion, results should be shortened to a single table fitting in a letter.

Remarks:

*Abstract structure is surprising: authors should expose the hypothesis more directly and clearly (e.g. in sentence 2 provides obvious idea, and last sentence seems tautological).  

*L49-50: antibodies directed against N protein are not generated by vaccination, but authors might rather say RNA vaccines were directed against S protein.

*L51- 54:  sentence should be abbreviated

*numbers should be truncated to 1 digit. 

*Table 1/2: check for commas.

*Table 2 check for legends.

*Table 3 seems useless, however it might be interesting to pool results obtained with each DMT in order to provide percent (sample) of not/vaccinated and COVID+/- at each visit. Percent of each DMT could be briefly given in text.

*Table should be referred to text (e.g. Table 6)

*Ref 26 is surprising: it was published before COVID19 and deals with a very different point.

Comments on the Quality of English Language

Manuscript should be deeply edited, including by a native english speaker

Reviewer 2 Report

Comments and Suggestions for Authors

The article “Analysis of the presence and levels of IgG antibodies directed against the S1 protein receptor binding subunit and the N protein of SARS-CoV2 in patients with multiple sclerosis treated with immunomodulatory therapies” is devoted to an important issue - the effect of receiving therapy in patients with multiple sclerosis on the level of antibodies against the spike protein (specifically the RBD domain) and N protein. The topic is important because immunomodulatory drugs can reduce the immune response, impairing the effectiveness of vaccines.

The authors conducted a fairly extensive experiment examining antibody levels to two targets in people (vaccinated and unvaccinated) with multiple sclerosis receiving immunomodulatory therapy. In general, the article contains extensive material and is provided with a sufficient number of tables.

However, I have some comments.

Minor:

1. The article contains a large number of spelling errors, although this is not so important.

2. I also believe that for the purpose of better understanding it would be possible to use not only tables but also illustrations; the material in the tables is not always easy to understand.

Major

I believe that the experimental plan was carried out incorrectly; the work does not include a group of vaccinated and unvaccinated people who do not have a diagnosis of multiple sclerosis. What is the actual difference between people with a diagnosis and receiving therapy from people without a diagnosis and, as a result, not receiving therapy?

Reviewer 3 Report

Comments and Suggestions for Authors

I was invited to revise the paper entitled "Analysis of the presence and levels of IgG antibodies directed against the S1 protein receptor binding subunit and the N protein of SARS-CoV2 in patients with multiple sclerosis treated with immunomodulatory therapies". It was a retrospective cohort tudy aimed to evaluate the IGG titers against sars-cov-2 proteins among MS patients. 

The topic is interesting and can help in imroving knowledge on this field.

Observations:

- The study was conducted in Poland, so Authors should report in introduction section how mass vaccination campign was developed in Poland;

- In table 2 Authors should improve presentation: type of vaccination should be added (MRNA based vaccine, viral vectore vaccine ecc.); time variable need unit of measurements; median has a typo (mediane);

- Why in 2023 in Poland did fragile patients received only two doses of vaccination?

- About statistical analysis, Wilcoxon test has to be used for paired data and not to comare two groups, as stated in lines 111;

- Line 120-121: Spearman tast evaluate the correlation and not the association;

- Authors reported to perform a GLM and a Mixed model but they should better present these results. In particular they should report the significance of each factor (gender, type of vaccination, drug therapy and covid19 infection). Table 10 is confusing;

Principal limitation of this study are:

- Lack in control group;

- Lack in type of vaccine used;

- Lack in drug therapy used during infection, such as monoclonal antinodies or antivirals;

- statistical analysis confused and poorly presented;

- history and type of MS disease can influence the study outcome.

Reviewer 4 Report

Comments and Suggestions for Authors

In this study, the authors examine the IgG production against two proteins of Sars-Cov2 in patients with multiple sclerosis. Some points must be addressed:

-      The treatment duration with DMT is not detailed.

-      In table 3, in the column “vaccinated/unvaccinated”, the authors should indicate the numbers of patients in each row.

-      In section 2.2, the abbreviations IgG-N and IgG-S1RBD are missing. Moreover, there is a mention of “IgA-N”, but no results are presented for IgA.

-      In table 6, a Fisher test was used but there is more than 2 groups, so a Chi 2 should be performed instead.

-      Finally, a global graphical representation of IgG levels as a function of time would be indicative for readers, with the 2 visits.
